# Cell Viability Assay and Surface Morphology Analysis of Carbonated Hydroxyapatite/Honeycomb/Titanium Alloy Coatings for Bone Implant Applications

**DOI:** 10.3390/bioengineering9070325

**Published:** 2022-07-18

**Authors:** Mona Sari, Ika Dewi Ana, Yusril Yusuf

**Affiliations:** 1Department of Physics, Faculty of Mathematics and Natural Science, Universitas Gadjah Mada, Yogyakarta 55281, Indonesia; monasari@mail.ugm.ac.id (M.S.); chotimah_w@ugm.ac.id (C.); 2Department of Dental Biomedical Sciences, Faculty of Dentistry, Universitas Gadjah Mada, Yogyakarta 55281, Indonesia; ikadewiana@ugm.ac.id

**Keywords:** carbonated hydroxyapatite, honeycomb, titanium alloy, electrophoretic deposition dip coating (EP2D), surface morphology, cell viability

## Abstract

In this work, carbonated hydroxyapatite/titanium alloy (CHA/Ti) and carbonated hydroxyapatite/honeycomb/titanium alloy (CHA/HCB/Ti) plates were coated using the electrophoretic deposition dip coating (EP2D) method. Analysis of cell viability and surface morphology of CHA/Ti and CHA/HCB/Ti coatings were carried out using the MTT (3-(4,5-Dimethylthiazol-2-yl)-2,5-diphenyltetrazolium bromide) assay and scanning electron microscopy (SEM), respectively. In a previous study, the thickness and average compressive strength values for the CHA/Ti and CHA/HCB/Ti plates were about 63–89 μm and 54–75 MPa, respectively. The result for thickness and compressive strength in this research followed the thickness and compressive strength parameters for coating in bone implants. In this work, the cell viability for incubation times during 24 h and 48 h of CHA/Ti plates is demonstrably superior to that of CHA/HCB/Ti plates, respectively, where the cell viability for CHA/Ti plates increased to ((67 ± 2)%) after incubation for 48 h. According to the one-way analysis of variance (ANOVA), the *p*-value was <0.05, indicating a significant difference in the average cell viability value across the three groups. Furthermore, the surface of CHA/Ti is not changed after the coating process. These results will yield many positive biomedical applications, especially in bone implants. Overall, CHA/Ti and CHA/HCB/Ti plates can be considered candidates for biomedical applications based on an analysis of surface morphology and cell viability.

## 1. Introduction

In the last decade, biomaterials have been used to regenerate damaged or diseased human body parts [1]. Metallic materials, especially stainless steel, magnesium-based alloys, cobalt-chromium alloy, and titanium and its alloys are widely used as medical implants, especially in dental and orthopedic surgeries [1,2,3,4]. Titanium (Ti) alloys have been popular for their corrosion resistance, mechanical properties, biocompatibility, low toxicity, wonderful antibacterial character, chemical stability, and superior photocatalysis [1,3,4,5,6,7]. Therefore, Ti is a candidate to become one of the most attractive biomaterials for bone implant procedures. Unfortunately, Ti alloys release aluminum and vanadium ions into body fluids under in vivo conditions that harm living systems [1], so the use of Ti alloys in bone implants affects pain and bruising in the surrounding tissue due to their low levels of biocompatibility [7]. Therefore, these alloys must be transformed using osteoconductive materials such as hydroxyapatite (HA) to coat the surface and improve osseointegration at the implant-bone tissue interfaces [7,8,9].

Carbonated hydroxyapatite (CHA) possesses better biological properties than HA for biomedical applications because of its low crystallinity and increased surface area [4]. This research used CHA based on biogenic materials from abalone mussel shells (*Haliotis asinina*) to coat Ti Alloy, as used in the previous study [7]. Abalone mussel shells from Indonesia contained high concentrations of calcium carbonate (CaCO_3_) as a natural precursor for the fabrication of CHA, which is 90–95% CaCO_3_ [10,11].

Biopolymers are progressively used in bone implant applications because of their chemical resemblances with organic tissue extracellular matrix (ECM) [12]. Therefore, much interest has concentrated on biopolymer coatings to refine the mechanical and biological behaviors of inorganic composite materials [3]. As a biopolymer [11], honeycomb (HCB) contextures achieve great strength with entirely interconnected pores of even size and high mechanical strength in the direction of the pores [7,13]. Still, HCB also contains proteins, vitamins, minerals, phytochemicals, and enzymes. However, the phytochemicals, phenolic acids, and flavonoids in HCB are considered beneficial to human health [14]. Therefore, in this research, Ti alloys were coated with a scaffold CHA/HCB 40 wt.%.

A variety of methods for coating metal surfaces has also been promoted, consisting of dip coating [15], electrophoretic deposition (EPD) [3,16,17], sol-gel coating [8,18], magnetron sputtering [19], and plasma spraying [20,21]. Among these, dip coating and EPD are the most convenient, but the EPD procedure demonstrates a greater cracking occasion because of manual withdrawal of the coating. Breaking in the layer dramatically influences the attributes when implanted in the body [22]. Furthermore, the dip-coating method requires more time [15]. Therefore, combining the two approaches controls surface morphology, free cracking adequately, and forming a homogeneous layer quickly. The combination of these methods produces an EP2D [7]. Electrophoretic deposition dip coater is a merger of a series of dip coater instruments and EPD tools that were integrated on the computer [7]. EPD and EP2D are differentiated based on their distinct coating processes. EPD only uses an electric field without a substrate withdrawal process, while EP2D uses an electric field and a substrate withdrawal process.

Cell viability is essential for analyzing a bio-composite material’s potential for biomedical applications, such as bone tissue engineering, bone implants, and dental implants. The most common method to measure cell viability in bio-composite materials is measuring metabolic activity by reducing MTT (3-(4,5-dimethylthiazole-2yl)-2,5-diphenyltetrazolium bromide). As a result, cellular enzymes, including mitochondrial NAD(P)H-dependent oxidoreductase and dehydrogenase, were reduced in populations exposed to toxic compounds [14]. Furthermore, the MTT assay was carried out to gauge the metabolic activity of MC3T3E1 cells and was estimated by the phenomenon of color change from yellow tetrazolium salt to purple formazan [11,23].

The preliminary study [7] demonstrated that CHA based on abalone mussel shells and scaffold CHA/HCB 40 wt.% was applied to coat the Ti alloy using the EP2D method. Based on preliminary research [11], CHA based on abalone mussel shells and scaffold CHA/HCB 40 wt% were demonstrated to be the best biomaterials by their physicochemical properties, including morphology, chemical composition, and crystallographic properties. Furthermore, the scaffold CHA/HCB 40 wt% had the potential scaffold for bone growth and cellular growth orientation because macropore and micropore sizes were ~102 ± 9.9 and ~1 μm, respectively. Overall, the FTIR spectra of scaffold CHA/HCB 40 wt% have presented the characteristic spectrum of CHA. In addition, the XRD pattern of the scaffold indicated the lower crystallinity, which must be lower because it affected dislocations, making it easier for cells to differentiate [11].

Previous studies [7] have also analyzed the physicochemical and mechanical properties of CHA/Ti and CHA/HCB/Ti layers. Moreover, the previous study revealed that the CHA/Ti and CHA/HCB/Ti plates with an immersion time of 30 min are the best sample based on physicochemical and mechanical properties analyses. Therefore, the novelties of this research aim to explore cell viability and surface morphology of samples using the MTT Assay and SEM analysis, respectively, with basic materials for CHA and scaffolds using Indonesian abalone mussel shells and honeycomb biopolymer, respectively. The study hypothesizes that adding CHA to CHA/Ti coating can increase osteoblast proliferation, and the incubation times on CHA/Ti and CHA/HCB/Ti plates have a significant effect on the ongoing proliferation process of MC3T3E1 cells.

## 2. Materials and Methods

The experimental procedure consisted of two main stages: the coating procedure of CHA/Ti and CHA/HCB/Ti plates using EP2D, with two titanium substrates used as the cathode and anode and characterization, including surface morphology and proliferation. The outline method for this study can be referred to in Figure 1.

### 2.1. Materials

The materials for this work, including CHA, a scaffold of CHA/HCB 40 wt%, CHA/Ti plates, and CHA/HCB/Ti plates to emulate conditions, were adapted from previous studies [7,10,11]. Penicillin-streptomycin, fungizone, and MEM-α medium were picked up from Gibco (Thermo Fisher Scientific, Carlsbad, CA, USA). Fetal bovine serum (FBS) and phosphate buffered saline (PBS) were bought from Sigma-Aldrich (Sigma-Aldrich Inc., Burlington, MO, USA). MTT was bought from Biobasic (Biobasic Inc., Amherst, NY, USA), and dimethyl sulfoxide (DMSO) was picked up from Merck (Merck KGaA, Darmstadt, Germany). Six-well plates were purchased from Nunc (Nalge Nunc International, Rochester, MA, USA).

### 2.2. Coating Method for CHA/Ti and CHA/HCB/Ti

According to methods demonstrated in previous research, processing for coating procedures, including Ti alloy substrate, CHA and scaffold CHA/HCB solutions, and coatings and calcination treatments for CHA/Ti and CHA/HCB/Ti were fabricated [7]. For the coating procedure, CHA/Ti and CHA/HCB/Ti plates were adjusted using EP2D, with two titanium substrates used as the cathode and anode. The CHA/Ti and CHA/HCB/Ti substrates were put into CHA/ethanol and CHA/HCB/ethanol, respectively, and were agitated with a magnetic stirrer (Thermo Fisher Scientific, Waltham, MA, USA) with a DC voltage of 50 V. The EPD handling was performed with a dyeing time of 30 min. The two substrates were attracted from the CHA/ethanol and CHA/HCB/ethanol solution at a velocity of 0.1 mm/s, controlled by a stepper motor by a computer. During the EPD process, CHA/ethanol and CHA/HCB/ethanol must be continually stirred to keep the stability of CHA/Ti and CHA/HCB/Ti coatings. The CHA/Ti and CHA/HCB/Ti were dried at room temperature before being calcinated at 900 °C for 3 h using a furnace (Vulcan, Yucaipa, CA, USA) [7].

### 2.3. Analysis of Compressive Strength for CHA/Ti and CHA/HCB/Ti

The compressive strength was measured using a Universal testing machine (TN20MD, Controlab, Paris, France). All compressive strength data were served as means ± standard deviation (SD) [24].

### 2.4. Analysis of Surface Morphology and Cell Viability Assay of CHA/Ti and CHA/HCB/Ti Plates

#### 2.4.1. Surface Morphology Analysis

The surface morphology of the plates was analyzed through SEM (JEOL JSM-6510LA-1400, Tokyo, Japan). First, all samples were embedded in a sample holder created from pure copper below vacuum to a pressure of 2.5 MPa and then coated with platinum by applying an ion coater. Next, the sample was incorporated into the SEM, and then an image pattern was produced in the form of the topography of the sample surface [24].

#### 2.4.2. Cell Viability Analysis

##### Cell Culture and Seeding

Pre-mouse osteoblasts (MC3T3EI) were cultured in MEM-α medium (Gibco, Carlsbad, CA, USA) + 10% fetal bovine serum (Gibco, Carlsbad, CA, USA) + 0.5% Fungizone (Gibco, Carlsbad, CA, USA). The state of the MC3T3E1 cells was observed with an inverted microscope. These cells can be harvested if they are 80% confluent. The media was discarded in the flask, and 3–4 mL of PBS 1× was poured into the flask. The flask was then closed and shaken to wash the cells from the remnants of the media. Then, PBS 1x was removed, and 0.5–1 mL of trypsin EDTA 0.25% (Gibco, CA, USA) was added and incubated for 4 min in a CO_2_ incubator. The flask was removed from the incubator and shaken to release the cells from the artificial flask matrix. The complete medium (10 mL) was added to inactivate trypsin EDTA 0.25%. The walls of the flask were rinsed to remove any adhering cells. The cells were resuspended and transferred to the sterile conical tube. The cells were centrifuged at 2500 rpm for 5 min, and the supernatant was discarded. The complete media (1 mL) was added and resuspended until homogeneous. Then, the cell suspension (10 µL) was taken, pipetted into a hemocytometer, and observed under a microscope. Finally, the cell count was completed using a counter. The CHA/Ti and CHA/HCB/Ti were added to the plate 6-well. A volume of 2 mL of the cellular suspension was seeded on the bottom of a 6-well plate at a density of 3 × 10^5^ cells/well. The cell-seeded on the plates were then incubated at 37 °C in 5% CO_2_ for 24 h and 48 h to investigate the proliferation activity of MC3T3E1 cells on all samples.

##### MTT Assay

The medium was discarded and 2 mL of MTT solution with a concentration of 0.5 mg/mL was added to the well and incubated for 4 h. Then, DMSO was added to the well at 2 mL/well. The absorbance was recorded by Tecan Spark^®^ (Tecan Trading AG, Männedorf, Switzerland) at 570 nm [11]. The following equation calculated the cell viability:(1)$$\mathrm{Cell}\;\mathrm{Viability}\;\left(\%\right)=\frac{\mathrm{absorbance}\;\mathrm{of}\;\mathrm{plates}-\mathrm{absorbance}\;\mathrm{of}\;\mathrm{control}\;\mathrm{media}}{\mathrm{absorbance}\;\mathrm{of}\;\mathrm{control}-\mathrm{absorbance}\;\mathrm{of}\;\mathrm{control}\;\mathrm{media}}\times 100$$

As shown in Equation (1), the analysis of cell viability was calculated based on the absorption percentage for unstimulated control cultures.

##### Statistical Analysis

All cell viability assay data were served as the mean ± standard deviation (SD), and one-way ANOVA was used to investigate the obtained results, followed by Tukey’s test to ensure that the data obtained were cell viability data. Tukey’s test was the multiple comparisons with the family-wised confidence level; however, each group was separated by the gradual density of factor components, and the maximum one obviously decreased. These data were statistically analyzed using OriginPro software version 2018 (OriginLab, Northampton, MA, USA) [24,25].

## 3. Results

### 3.1. Physicochemical and Mechanical Properties of CHA/Ti and CHA/HCB/Ti Plates

The physicochemical properties of CHA/Ti and CHA/HCB/Ti plates have been characterized through cross-section and crystallographic properties and mechanical properties through compressive strength analysis, as shown in the previous study [7]. Based on Table 1 and Figure 2a–d, the average thickness and compressive strength values of the CHA/HCB/Ti plate are more significant than those of the CHA/Ti plate because the CHA/HCB layer deposited on the Ti surface is thicker than the CHA/Ti surface. Additionally, because CHA/HCB requires a longer suspension time than CHA alone, this method generates a more significant number of samples that are too thick to be used. The thickness value for the sample was about 63–88 μm. In this study, the compressive strength value for the control group (Ti alloy) was higher than the experimental group (Figure 2c). However, during the testing process, the CHA/Ti and scaffold CHA/HCB 40 wt.% was broken after pressure from the compressive strength testing tool, reducing its compressive strength [7]. In this study, the compressive strength for all immersion time variations (Figure 2c) was about 54–83 MPa, which was within the normal range of human cancellous bone (0.2–80 MPa) [26].

The X-ray diffractometer (XRD) pattern exhibited the HA, CHA, and Ti phases (Figure 2d). The peak of the CHA phase was more intense for the CHA/HCB/Ti plate than for the CHA/Ti plate. Since the calcination process is performed in a normal atmosphere, most Ti peaks disappeared. Instead, characteristic peaks of rutile TiO_2_ were exhibited [4]. The CHA/Ti plate showed a more significant decrease in crystallite size and increased microstrain than the CHA/HCB/Ti plate. This phenomenon is thought to be connected to the intensity of the Ti alloy phase, which was still less intense, and the presence of the HA phase experienced by the CHA/Ti plate. Moreover, changes in lattice parameters developed because of the calcination treatment, which shrinks the lattice in CHA and CHA/HCB [7].

### 3.2. Surface Morphology and Cell Viability Assay Analysis of CHA/Ti and CHA/HCB/Ti Plates

The surface morphology and cell viability analysis of the plates can be seen in Figure 3a–c. CHA/Ti coating formed uniform surface morphology and showed a high aggregation of CHA particles, as shown in Figure 3a. Figure 3b indicated that CHA/HCB/Ti coatings developed the honeycomb pore structure on its surface morphology, produced from HCB in the CHA/HCB bio-composite scaffold [7]. As shown in Figure 3c and Table 2, the cell viability of CHA/Ti is superior when compared to the plates with CHA/HCB/Ti coatings. This is also supported by layer analysis of the CHA/Ti. The cell viability increased to 53–67% after incubation for 48 h. CHA is also known to increase osteoblast proliferation because of the carbonate content [27,28,29]. Therefore, CHA/Ti plates can promote cell proliferation. Based on the one-way ANOVA, the *p*-value was <0.05, indicating a significant distinction in the average cell viability value across the three groups. This means that the incubation times of 24 and 48 h on CHA/Ti and CHA/HCB/Ti plates affected the ongoing proliferation process of MC3T3E1 cells.

## 4. Discussion

The development of EP2D is anticipated to control surface morphology, layer thickness, free cracking, and so on [7]. These results can be identified through the surface morphology of CHA/Ti and CHA/HCB/Ti plates. Adding HCB to the CHA/Ti coating and applying a fixed voltage of 50 V can reduce cracking in the surface morphology of samples. Furthermore, the higher calcination temperature in the coating process affected stronger adhesion of the coating to the substrate due to the diffusion effect [22]. The use of the EP2D method during the coating process was similar to the Plasma electrolytic oxidation (PEO) procedure. However, applying the PEO method to the incorporations of Zn and Mg elements into the oxide layer of Tie6Ale4V alloy can result in cracked pores [30]. This result was the same as this research, in that the addition of HCB biopolymer on the surface of the CHA/Ti layer with the EP2D method resulted in cracked pores. Cracks in the sample were due to the rapid solidification of the molten material [31].

Reduced CHA/HCB/Ti cracking also caused by higher stress will accelerate deposition time due to the influence of a more vital electrostatic force [22]. Thus, the adhering particles will become denser. Overall, the small withdrawal speed (0.1 mm/s) and high stress (50 V) affect aggregation on the surface morphology so that cracking is reduced. High withdrawal speeds can result in the deposition of particles that have been shaken and released. The value of the binding energy that occurs is lower than the energy withdrawn.

The HCB contained alkanes chains that had not evaporated on the scaffold fabrication perfectly. Alkanes chains caused damage to pre-osteoblast MC3T3E1 cells [32], so the percentage of cell viability assay of CHA/HCB/Ti was less than CHA/Ti, as shown in Figure 3c. This result contrasts with the effects of TiO_2_ particles on the viability of cells after 24 and 48 h. Cell viability with an incubation time of 48 h was significantly reduced in a concentration-dependent manner by TiO_2_ exposure. Cell viability was not changed at the lower concentration (1 mg/L) compared to control data, same as using Vive Nano Titania; the cell viability remained unchanged at 1 and 10 mg/L. Still, it was decreased to 85% of the control at 100 mg/L after incubation time for 48 h [33]. Moreover, the cell viability of CHA/Ti is demonstrably better than CHA/HCB/Ti plates, even though the CHA/Ti plates were also the thinner of the samples measured. This result is beneficial because the CHA/Ti surface does not need to be changed after the coating process. Moreover, calcium salts and phosphorus salts as electrolytes are standard methods to improve the viability cell of titanium alloys in the PEO process [34], as is the addition of CHA to CHA/Ti and CHA/HCB/Ti coatings using the EP2D method. Therefore, CHA, one example of CaP, may promote the proliferation of osteoblasts to some extent [35].

The lower mechanical properties of CHA/Ti can be caused by technical factors; namely, the coating layer broke after experiencing pressure from the compressive strength test equipment and the layer composition factor [2], including the content of the Ti alloy and the CHA decomposition process itself. However, mechanical properties, such as compressive strength, have no significant effect on the success rate of implants. Compression tests are often performed to assess biomechanical properties, for example, on full-thickness articular cartilage. However, little is known about the viability of articular cartilage cells under constant mechanical stress that would compress the ECM similarly to the usual conditions experienced by joints [36]. Thus, compressive strength did not affect the success rate of the implant.

This study has some limitations due to principal and technical challenges. Further experiments should be carried out to refine the findings of this research in the future, such as comprehensive mechanical properties through bonding strength. In addition, biocompatibility through contact angle, in vitro biocompatibility including cell adhesion, assessment in simulated body fluid (SBF), and in vivo tests for all coating samples for clinical practice should also be studied.

## 5. Conclusions

This work is a productive characterization and analysis of CHA/Ti and CHA/HCB/Ti coatings through surface morphology and cell viability assay analysis. The hypothesis is that adding CHA to CHA/Ti coating can increase osteoblast proliferation, and the incubation times on CHA/Ti and CHA/HCB/Ti plates significantly affected the ongoing proliferation process of MC3T3E1 cells has been accepted. Furthermore, the cell viability value of CHA/Ti is greater than that of CHA/HCB/Ti coatings, where the cell viability for CHA/Ti plat increased to ((67 ± 2)%) after incubation for 48 h. The percentage of the cell viability assay of CHA/HCB/Ti was less than CHA/Ti because the HCB contained alkane chains that had not evaporated on the scaffold fabrication perfectly. Alkane chains caused damage to pre-osteoblast MC3T3E1 cells. In addition, CHA/Ti demonstrates a thinner layer on its surface, which is favorable because the CHA/Ti surface remains unchanged after the coating process.

## Figures and Tables

**Figure 1 bioengineering-09-00325-f001:**
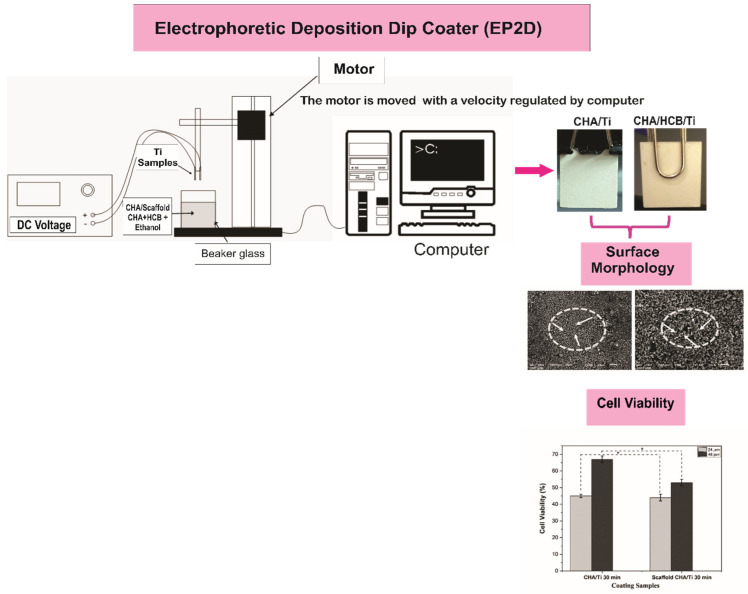
The schematic design used for the coating procedures (modification from [7]).

**Figure 2 bioengineering-09-00325-f002:**
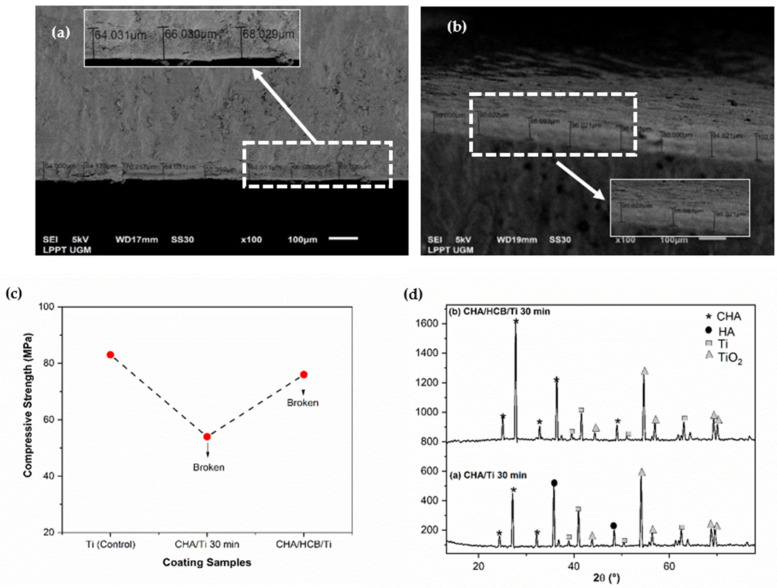
Physicochemical and mechanical properties of coating samples. (**a**) Cross-section of CHA/Ti 30 min, (**b**) cross-section of CHA/HCB/Ti 30 min, (**c**) compressive strength graph, and (**d**) crystallographic properties of samples [7].

**Figure 3 bioengineering-09-00325-f003:**
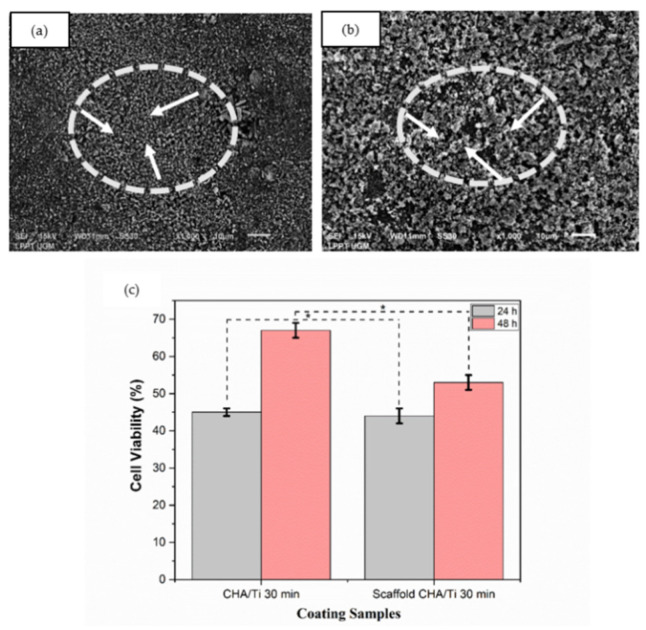
Surface morphology of coatings with an immersion time of 30 min. (**a**) CHA/Ti and (**b**) CHA/HCB/Ti (the white arrows indicate surface morphology), and (**c**) cell viability assay of coating samples after being incubated for 24 h and 48 h (*: *p* < 0.05).

**Table 1 bioengineering-09-00325-t001:** Physicochemical and mechanical properties of CHA/Ti 30 min and CHA/HCB/Ti 30 min samples [7].

No.	Coating Samples	Average Thickness Value (μm)	Crystallographic Properties					Compressive Strength (MPa)
			Crystallite Size (nm)	Microstrain	Lattice Parameter (Å)			
					*a*	*c*	*a/c*	
1	CHA/Ti	63 ± 6	18 ± 3	0.0042	9.63	7.29	0.76	54
2	CHA/HCB/Ti	89 ± 6	20 ± 2	0.0070	9.50	7.11	0.74	76

**Table 2 bioengineering-09-00325-t002:** Average Cell Viability of CHA/Ti 30 min and CHA/HCB/Ti 30 min.

No.	Coating Samples	Cell Viability (%)		*p*-Value
		Mean ± SD		
		24 h	48 h	0.000
1	Control (Ti)		27 ± 0.2	
2	CHA/Ti	45 ± 1	67 ± 2	
3	CHA/HCB/Ti	44 ± 2	53 ± 2	

## Data Availability

The data presented in this study are available on request from the corresponding author.

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
