# Peer review of "Cell Viability Assay and Surface Morphology Analysis of Carbonated Hydroxyapatite/Honeycomb/Titanium Alloy Coatings for Bone Implant Applications"

_bioengineering, 2022, doi:10.3390/bioengineering9070325_

Round 1

Reviewer 1 Report

Dear authors,

Your article is interesting and it is worth publishing it. Brings new information to implant dentistry field. Some changes are needed as follows:

1) You entitle the type of manuscript as "communication"; it should be "article" or original research, it depends how it is written in the journal perspective. 

2) Abstract: many language mistakes and grammar. Please revise it carefully.

3) Line 65: You should mentioned something about the use of drugs and systemic diseases because these factors are quite important in implant osseointegration. This articles may help you (doi: 10.3390/medicina58030343; doi: 10.1111/clr.13602; doi: 10.1080/03602532.2019.1687511)

4) Line 97: Write what is the clear aim of your study.

5) Line 105: describe both methods.

6) Line 183: Statistical analysis should be in a separate chapter/subchapter.

7) Discussion chapter should be developed more. You should make comparisons with other study. There is too low information.

8) Conclusion should be written in a clearer way.

Author Response

Prof. Dr. Anthony Guiseppi-Elie 

Editor-in-Chief

  1. Founding Dean, College of Engineering, Anderson University, Anderson, SC 29621, USA
  2.  Department of Cardiovascular Sciences, Houston Methodist Institute for Academic Medicine, Houston Methodist Research Institute, 6670 Bertner Ave., Houston, TX 77030, USA

Dr. Zhong Zheng 

UCLA School of Dentistry,10833 Le Conte Ave.Box 951668, Los Angeles, CA 90095-1668, USA
Guest Editor

Special Issue Bioengineering "New Sight of Biomaterials and Tissue Regeneration in Medicine Research: Updates and Future Direction

Dear Prof. Dr. Anthony Guiseppi-Elie and Dr. Zhong Zheng 

The authors would like to thank and express our appreciation to the editor and reviewers for allowing us to revise our manuscript. The authors received many constructive recommendations to improve the manuscript. We are pleased to resubmit our new version of the paper entitled "Cell Viability Assay and Surface Morphology Analysis of Carbonated Hydroxyapatite/Honeycomb/Titanium Alloy Coatings for Biomedical Application" by Mona Sari, Chotimah, Ika Dewi Ana, and Yusril Yusuf for consideration of publication in the Special Issue Bioengineering "New Sight of Implant and Bone Regeneration in Medicine Research: Updates and Future Direction

Our responses to the editor and reviewer comments are given below.

Reviewer 1 Comments

Point 1: You entitle the type of manuscript as "communication"; it should be "article" or original research, it depends how it is written in the journal perspective.

Response 1: The type of this article has been changed to be the article.

Point2: Abstract: many language mistakes and grammar. Please revise it carefully.

Response 2: The authors have fully improved the grammatical expression in this manuscript (red word/phrase), especially for the abstract, page 1. English corrections have been checked using Grammarly Applications ( https://app.grammarly.com/).

Point 3: Line 65: You should mentioned something about the use of drugs and systemic diseases because these factors are quite important in implant osseointegration. This articles may help you (doi: 10.3390/medicina58030343; doi: 10.1111/clr.13602; doi: 10.1080/03602532.2019.1687511)

Response: In an introduction section regarding osseointegration, the authors only mention that Ti alloys release aluminum and vanadium ions into body fluids under in vivo conditions that harm living systems. Therefore, these alloys must be coated using osteoconductive materials such as HA to cover the surface and improve osseointegration at the implant-bone tissue interfaces. So, they only give the example of osteoconductive materials to enhance osteointegration Ti. Unfortunately, they don't explain something about the use of drugs and systemic diseases because these factors are quite important in implant osseointegration, so we can't cite these articles.

Point 4: Line 97: Write what is the clear aim of your study.

Response: The clear aim and hypothesis of this study have been added in the last paragraph of the introduction, lines 115–118 on page 3.

Point 5: Line 105: describe both methods.

Response: The description of the coating procedure and this characterization has been written in sections 2.2–2.4 on pages 5–6.

Point 6: Line 183: Statistical analysis should be in a separate chapter/subchapter.

Response: Statistical analysis has been separated in section 2.4.2.3, page 6.

Point 7: Discussion chapter should be developed more. You should make comparisons with other study. There is too low information.

Response: The discussion has been revised with the addition of the comparison with other studies in the discussion section on page 9 that can show in References 30,31, 33–35.

Point 8: Conclusion should be written in a clearer way.

Response: The conclusion writing has been revised, pages 10–11.

Sincerely,

Mona Sari

Chotimah

Ika Dewi Ana

Yusril Yusuf (corresponding author)

Reviewer 2 Report

This work examined the cell viability and surface morphology of carbonated hydroxyapatite/honeycomb/titanium alloy coatings for bone implant applications. The manuscript can be accepted for publication in Bioengineering after major revision.

1.  It is suggested to present more quantitative data in the abstract.

2. The introduction part should be improved by shedding the light on the novelty of the present work.

3. Please index all peaks in XRD patterns

4. Fig.3c has not appeared properly. Please provide a better image.

5. The scale bars in Fig.3 are not readable.

6. The conclusion must be revised by adding more quantitative results. The present conclusion is very short.

7. The captions to the figures must give full details so that the information can be clearly understood by the reader

8. The relevant literature should be cited in the manuscript. The results can be compared with other surface modification methods. The following examples are highly recommended:

(a) https://doi.org/10.3389/fmats.2022.883027

(b) https://doi.org/10.1016/j.jallcom.2020.156840

(c) https://doi.org/10.1016/j.jallcom.2019.153038

(d) https://doi.org/10.1016/j.matdes.2020.108758

(e) https://doi.org/10.3390/met11010106

(f)  10.1016/j.tiv.2012.12.018

(g) 10.1039/D0TB00697A

Author Response

Prof. Dr. Anthony Guiseppi-Elie 

Editor-in-Chief

  1. Founding Dean, College of Engineering, Anderson University, Anderson, SC 29621, USA
  2.  Department of Cardiovascular Sciences, Houston Methodist Institute for Academic Medicine, Houston Methodist Research Institute, 6670 Bertner Ave., Houston, TX 77030, USA

Dr. Zhong Zheng 

UCLA School of Dentistry,10833 Le Conte Ave.Box 951668, Los Angeles, CA 90095-1668, USA
Guest Editor

Special Issue Bioengineering "New Sight of Biomaterials and Tissue Regeneration in Medicine Research: Updates and Future Direction

Dear Prof. Dr. Anthony Guiseppi-Elie and Dr. Zhong Zheng 

The authors would like to thank and express our appreciation to the editor and reviewers for allowing us to revise our manuscript. The authors received many constructive recommendations to improve the manuscript. We are pleased to resubmit our new version of the paper entitled "Cell Viability Assay and Surface Morphology Analysis of Carbonated Hydroxyapatite/Honeycomb/Titanium Alloy Coatings for Biomedical Application" by Mona Sari, Chotimah, Ika Dewi Ana, and Yusril Yusuf for consideration of publication in the Special Issue Bioengineering "New Sight of Implant and Bone Regeneration in Medicine Research: Updates and Future Direction

Our responses to the editor and reviewer comments are given below.

Reviewer 2 Comments

Point 1: It is suggested to present more quantitative data in the abstract.

Response: The quantitative data has been shown in the abstract, page 1.

Point 2: The introduction part should be improved by shedding the light on the novelty of the present work.

Response: The explanation about the novelty in this work has been added in the last part of the introduction, pages 3 and 4.

Point 3: Please index all peaks in XRD patterns

Response: The addition indexed some peaks in the XRD pattern has been added in Fig 2 (d), page 7.

Point 4: Fig.3c has not appeared properly. Please provide a better image

Response: Figure 3c has been revised, page 9.

Point 5: The scale bars in Fig.3 are not readable

Response: The showing of scale bars in Fig.3 (a) and (b) has been revised, page 9.

Point 6:  The conclusion must be revised by adding more quantitative results. The present conclusion is very short.

Response: The conclusion has been revised by adding the quantitative data about cell viability, pages 10 and 11.

Point 7: The captions to the figures must give full details so that the information can be clearly understood by the reader.

Response: The detailed information of the captions to the Figures has been revised, especially in Figure 2.

Point 8: The relevant literature should be cited in the manuscript. The results can be compared with other surface modification methods. The following examples are highly recommended:

(a) https://doi.org/10.3389/fmats.2022.883027

(b) https://doi.org/10.1016/j.jallcom.2020.156840

(c) https://doi.org/10.1016/j.jallcom.2019.153038

(d) https://doi.org/10.1016/j.matdes.2020.108758

(e) https://doi.org/10.3390/met11010106

(f)  10.1016/j.tiv.2012.12.018

(g) 10.1039/D0TB00697A

Response: The results have been revised with the addition of the comparation with other surface modification methods, cited in references c, d, and for numbers 30,33,35 in this manuscript as recommendation reviewers and the authors have added other references number 31 and 34, pages 9 and 10.

Reviewer 3 Report

This paper is a continuation of two previous works of the authors [1-2]. An electrophoretic deposition dip coating (EP2D) method was used to prepare the previously reported coatings. 

[1] Porous structure of bioceramics carbonated hydroxyapatite-based honeycomb scaffold for bone tissue engineering

[2] Carbonated Hydroxyapatite-Based Honeycomb Scaffold Coatings on a Titanium Alloy for Bone Implant Application—Physicochemical and Mechanical Properties Analysis 

Although the authors have previously researched coatings prepared by the EP2D method, previously researched have also shown that this method can be used to prepare CHA/HCB/Ti coatings. However, the work in this paper is only a continuation of the previous work and only some biological properties were tested. The authors are honest about the limitations of this study (4. Discussion Lines: 311-317) and intend to continue with in vitro biocompatibility tests and SBF evaluation (in vitro bioactivity) tests in subsequent studies. Therefore, this paper lacks innovation. Overall, the value of the authors' work cannot be denied.

Before the paper can be published, several issues need to be addressed.

(1)1.Introduction

Lines: 64-65. Why was 40 wt.% HCB selected for the study? What were the reasons?

(2)2.4.2.1 Cell culture and seeding

Lines: 177-178. Why are cells inoculated on the bottom of the culture plate instead of the coating surface? What is the purpose of doing this?

(3)2.4.2.2 MTT Assay and its staticstical analysis

How to ensure that the test data is the cell viability on the coating?

(4)5. Conclusions

Since the authors' study found that the cell viability value of CHA/Ti coating is greater than that of CHA/HCB/Ti coating, why is it necessary to add HCB to prepare the coating? Moreover, in 1. Introduction (Lines: 55-65), it is described that HCB seems to be beneficial to the organism. Please comment on the possible reasons why the cell viability value of CHA/Ti coating is greater than that of CHA/HCB/Ti coating.

Author Response

Prof. Dr. Anthony Guiseppi-Elie 

Editor-in-Chief

  1. Founding Dean, College of Engineering, Anderson University, Anderson, SC 29621, USA
  2. 2. Department of Cardiovascular Sciences, Houston Methodist Institute for Academic Medicine, Houston Methodist Research Institute, 6670 Bertner Ave., Houston, TX 77030, USA

Dr. Zhong Zheng 

UCLA School of Dentistry,10833 Le Conte Ave.Box 951668, Los Angeles, CA 90095-1668, USA
Guest Editor

Special Issue Bioengineering "New Sight of Biomaterials and Tissue Regeneration in Medicine Research: Updates and Future Direction

Dear Prof. Dr. Anthony Guiseppi-Elie and Dr. Zhong Zheng 

The authors would like to thank and express our appreciation to the editor and reviewers for allowing us to revise our manuscript. The authors received many constructive recommendations to improve the manuscript. We are pleased to resubmit our new version of the paper entitled "Cell Viability Assay and Surface Morphology Analysis of Carbonated Hydroxyapatite/Honeycomb/Titanium Alloy Coatings for Biomedical Application" by Mona Sari, Chotimah, Ika Dewi Ana, and Yusril Yusuf for consideration of publication in the Special Issue Bioengineering "New Sight of Implant and Bone Regeneration in Medicine Research: Updates and Future Direction

Our responses to the editor and reviewer comments are given below.

Reviewer 3 Comments

Point 1: Introduction

Lines: 64-65. Why was 40 HCB selected for the study? What were the reasons?

Response: The scaffold CHA/HCB 40  had the potential scaffold for bone growth and cellular growth orientation because macropore and micropore sizes were  102  9.9 and  , respectively. Overall, the FTIR spectra of scaffold CHA/HCB 40  have shown the characteristic spectrum of CHA. In addition, the XRD pattern of the scaffold indicated the lower crystallinity, which must be lower because it affected dislocations, making it easier for cells to differentiate. This explanation about scaffold CHA/HCB 40  selected for coating with Ti alloy using EP2D method has been added in the introduction, page 3.

Point 2: 2.4.2.1 Cell culture and seeding

Lines: 177-178. Why are cells inoculated on the bottom of the culture plate instead of the coating surface? What is the purpose of doing this?

Response: For technical reasons, MC3T3EI cells will be seeded in a 6-well plate which has been added with CHA/Ti and CHA/HCB/Ti into the 6-well plate to investigate the proliferation activity of MC3T3E1 cells on all samples.

Point 3: 2.4.2.2 MTT Assay and its statistical analysis

How to ensure that the test data is the cell viability on the coating?

Response:  All cell viability assay data were served as the mean standard deviation (SD), and one-way ANOVA was used to investigate the obtained results, followed by Tukey's test to ensure that the data obtained were cell viability data. Tukey's test was the multiple comparisons with the family-wised confidence level. Tukey's test was the multiple comparisons with the family-wised confidence level. But each group was separated by the gradual density of factor components, and the maximum one obviously decreased. These data were statistically analyzed using OriginPro software version 2018 (OriginLab, Northampton, MA, USA). This explanation has been added in the statistical analysis section, page 6.

Point 4: 5. Conclusions

Since the authors' study found that the cell viability value of CHA/Ti coating is greater than that of CHA/HCB/Ti coating, why is it necessary to add HCB to prepare the coating? Moreover, in 1. Introduction (Lines: 55-65), it is described that HCB seems to be beneficial to the organism. Please comment on the possible reasons why the cell viability value of CHA/Ti coating is greater than that of CHA/HCB/Ti coating.

Response: The percentage of cell viability assay of CHA/HCB/Ti was less than CHA/Ti because the HCB contained alkanes chains that had not evaporated on the scaffold fabrication perfectly. Alkanes chains caused damage to Pre-Osteoblast MC3T3E1 cells. This explanation has been added in conclusion, pages 10–11.

Sincerely,

Mona Sari

Chotimah

Ika Dewi Ana

Yusril Yusuf (corresponding author)

Round 2

Reviewer 2 Report

The given comments are fully done. It can be accepted in present form